# Backward Energy Transmission in Resonant RF Energy Harvesters

**DOI:** 10.3390/mi16101187

**Published:** 2025-10-21

**Authors:** Jakub Szut, Mariusz Pauluk, Paweł Piątek

**Affiliations:** Faculty of Electrical Engineering, Automatics, Computer Science, and Biomedical Engineering, Department of Automatic Control and Robotics, AGH University of Krakow, al. A. Mickiewicza 30, 30-059 Krakow, Poland; mp@agh.edu.pl (M.P.); ppi@agh.edu.pl (P.P.)

**Keywords:** RFEH, resonance, energy transfer, backward propagation, simulations

## Abstract

RF Energy Harvesting (RFEH) circuits have been extensively researched in recent years. Researchers have proposed dozens of RFEH models with various architectures and topologies. Due to the small amount of energy available to be harvested, RFEH circuits must be as efficient as possible, both in terms of receiving energy and its further processing. Recent research has identified that in resonant circuits, some of the received energy is retransmitted and therefore lost.

## 1. Introduction

Energy consumption around the world is increasing every year [1]. The global trend is to reduce the use of fossil fuels in order to protect the environment. Although more efficient energy sources are used for general purposes, there are certain applications where power requirements are much lower and can be met using unconventional energy sources. Recent developments in electronics have led to the miniaturization of thousands of devices. This has opened up many new use cases in various fields, including the Internet of Things (IoT) [2]. Currently, connected sensors are used in a wide range of environments. From industry and transportation to agriculture and households, these sensors monitor and provide essential data while controlling critical processes. The rapid increase in the number of IoT sensors has required the development of new communication and power supply methods. Purpose-driven communication was already addressed in the fifth-generation (5G) telecommunication standard through Massive Machine-Type Communication (mMTC) [3]. Despite some progress in optimizing energy consumption, it is the sixth generation (6G) that envisions so-called “zero energy” sensors [4]. These microdevices are designed to operate with only minimal energy consumption [5]. Since using batteries or installing power lines for hundreds of sensors is suboptimal, recent developments in energy harvesting have introduced new ways to power these devices [6]. Several energy sources are used in harvesting techniques, including well-known ones such as solar, wind, vibration, and thermal sources. Moreover, research into new forms of renewable energy has revisited an idea first proposed more than a century ago by Nikola Tesla [7]. Tesla’s original concept has since evolved, shifting the focus from energy transmission to energy harvesting. This evolution has led to the development of Radio Frequency Energy Harvesters (RFEH) [8]. Because RF transmissions provide only very small amounts of energy, minimizing losses is critical. Therefore, the design and efficiency of harvesters play a central role. These microdevices must collect, rectify, and stabilize the energy received from RF transmissions. Although the precision of electronic modules continues to improve, some issues remain. One of the main challenges identified in recent research is backward energy transmission in resonant RF energy harvesters. In this paper, the general concept of Radio Frequency Energy Harvesters is briefly presented in Section 2. Various technologies and topologies of such devices are introduced in Section 3, an antenna model and an analytical model of dipole impedance are provided in Section 4 and Section 5, respectively, and the issue of backward energy transmission is examined using simulation models in Section 6. A potential solution to the problem of backward energy transmission is proposed, and the corresponding results are simulated in Section 7 and discussed in Section 8. The research presented is concluded in Section 9. Finally, in Section 10, the directions for future work are outlined.

## 2. Radio Frequency Energy Harvesters

Some amount of energy is constantly transmitted by wireless communication systems via radios [9]. Due to the small amount of energy available during radio transmission, solutions using dedicated transmitters that enable wireless energy transfer have been the most popular [10]. However, the higher bandwidth of telecommunications networks, as well as the use of increasingly wider millimeter wave bands, has allowed us to turn our attention to harvesting the energy emitted during the operation of systems such as mobile networks and Internet access points. However, the amount of available energy is still small and requires processing in a way that minimizes losses [11]. In general, RF Energy Harvester is a device consisting of a few stages [12]. The first is the receiving antenna. This part is often designed to receive as much energy as possible at a given frequency or in some frequency range [13]. Just after the antenna, there is an impedance matching network. This part is crucial in minimizing losses during energy transfer from the antenna to the rest of the harvester module. Through an impedance matching network, energy is transferred to the rectifier. The energy received on the antenna is in the form of alternating current (AC); It has to be rectified to be used in the target application such as an IoT sensor. As the amount of energy provided by the antenna is often very small, there is a voltage multiplier, which can even have a few stages [14]. Without a proper voltage level, the harvested energy cannot be used to power the target device. This is how the basic RF Energy Harvester is designed. Very often it also has some power managements unit that is controlling the output of the harvester and how the available energy is applied further. The crucial aspect of the design of the RF Energy Harvesters is the amount of energy available to be harvested. These devices have to be as efficient as possible to prevent any losses of already small amounts of energy. New electronic components with increasingly better parameters make it possible to achieve this goal, but it is necessary to design the energy conversion system appropriately for low power and high frequencies. Only then is it possible to convert the energy harvested into a form that can power the end system. Although such devices may seem to lack potential use cases in real-life scenarios, there are a few commercial RF Energy Harvesters [15], such as the Belgian company e-peas harvester AEM30940 working on the 915 MHz band which was presented on Figure 1.

## 3. Technologies and Topologies

In order to meet the requirements of high-performance systems, researchers in the field of electromagnetic wave energy harvesting are continuously developing new technologies and topologies for energy harvesting systems. Several main research directions can be identified in the literature. The most widely used and also the most mature technology is based on Schottky diodes [16]. These diodes provide low forward voltage drop and short switching times, making them suitable for high-frequency systems. A more recent approach involves the use of transistors [17]. Advances in semiconductor manufacturing have significantly improved the efficiency of these components, and CMOS technology is increasingly being used in the design of electromagnetic wave energy harvesting systems [18]. Another area of research focuses on circuit topology. Early designs relied on half- or full-wave rectifier configurations [19]. Later, these were extended to models that incorporate voltage multipliers, such as Cockcroft–Walton multipliers [20]. This topology increases the output voltage of the harvesting system, allowing the harvested energy to power additional segments of the circuit. A topology that has recently gained popularity is the Coupled Cross Differential Drive (CCDD) [21]. Regardless of the technology or topology employed, a critical component is the matching network between the receiving antenna—where the transmitted energy is captured—and the rest of the processing system. Matching networks are designed to ensure optimal conditions for lossless energy transfer from the antenna to the circuit. Furthermore, since resonance may occur in such systems, an appropriate matching network helps to reduce energy flow losses. There are a few approaches to matching network design [22]. The design is often module-specific and modeled for the targeted frequency. Hence, proposing a wideband harvester is not trivial, as it requires a specific approach to the matching network [23]. Although matching networks are important, it has been observed that some energy loss is still present in the system. It is difficult to follow them as research has not yet been conducted under uniform conditions to collect and compare data. However, it is postulated to standardize the research on RF Energy Harvesters [24]. Although many sources of losses are described in the literature, this research presents the one which comes from the energy transfer between the receiving antenna and the rest of the circuit.

## 4. Antenna Model

We assume that the antenna is a circuit of mixed resistance and reactance [25] with the impedance: $Z_{ant}$ is generally described as follows:(1)$$Z_{ant}=R_{ant}+jX_{ant}$$
where
$Z_{ant}$—impedance of an antenna;$R_{ant}$—resistance of an antenna;$X_{ant}$—reactance of an antenna.

The antenna resistance $R_{ant}$ can also be considered as the radiation resistance $R_{rad}$ and the loss resistance $R_{loss}$, and the reactance as the inductance $L_{ant}$ and the capacitance $C_{ant}$. The radiation resistance models the effect of changing electric energy in electromagnetic radiation. The loss resistance is directly equal to the electric resistance of the material in which the antenna is built. Formally, this is as follows:(2)$$R_{ant}=R_{rad}+R_{loss}$$
with
$R_{rad}$—radiation resistance;$R_{loss}$—loss resistance.
(3)$$X_{ant}=jX_{Lant}-jX_{Cant}$$
where
$X_{Lant}$—reactance as antenna’s inductance;$X_{Cant}$—reactance as antenna’s capacitance.

For the signal with characteristic frequency, for which the antenna is designed: $X_{ant}=0$, and the main part of the energy received by the antenna is radiated, and the rest is dissipated in the loss resistance, since $R_{loss}<<R_{rad}$ is usually. The reactants on the right side of (Equation 3), as shown in the text, are not constant and vary in frequency references.

## 5. Analytical Model of the Dipole Impedance

Let us assume the following:*l*—length of the dipole;λ—length of the wave;$e_{l}$—electrical length;*d*—radius of the rod, the dipole is constructed.

Electrical length $e_{l}=l\cdot 2\pi /\lambda$. In practice, the result of this product is a universal parameter for every antenna, expressed in radians and equal to 2π when the dipole length is equal to the length of the wave.

Considering the dipole antenna, one may find, e.g., in [26,27] analytical equations for impedance approximated with a third-order polynomial; see (Equation 4)–(Equation 6).(4)$$Z_{ant}\left(e_{l}\right)=R_{ant}\left(e_{l}\right)-j\left[120\left(\ln\frac{l}{d}-1\right)\cot e_{l}-X_{ant}\left(e_{l}\right)\right]$$(5)$$R_{ant}\left(e_{l}\right)\approx -0.4787+7.3246e_{l}+0.3963{e_{l}}^{2}+15.6131e_{l}^{3}$$(6)$$X_{ant}\left(e_{l}\right)\approx -0.4456+17.0082e_{l}-8.6793e_{l}^{2}+9.6031e_{l}^{3}$$

The above formulas reflect the behavior of the antenna impedance only in the vicinity of the series resonance point [26]. This is satisfying in many applications. Currently, when analytical formulas are too complicated to solve or the problem is too sophisticated to find adequate equations, numerical methods play a significant role. In the following, an example of the dipole antenna impedance calculation is given with reference to frequency (Figure 2). The dipole is metal, made of cylindrical rods. The diameter of the rod is d=1 mm and the length of the dipole is equal to l=0.0625 mm (both wings), which corresponds to wavelength λ=0.125 mm and resonance frequency $f_{res}=2.4$ GHz. The impedance is calculated in the MATLAB 2025a antenna toolbox, which uses the method of moments computation techniques for metal surfaces. Calculations are made for a 4 mm dipole strip model, equivalent to a cylindrical antenna with 1 mm diameter [28].

According to the calculations above, the resonance frequency is approximately equal to 2.2 GHz, where $X_{ant}=0\;\Omega$ and $R_{ant}\approx 70\;\Omega$, which occurs below the designed 2.4 GHz. The simulation shows that the antenna dimensions need to be adjusted to achieve the required resonance frequency. Both Equations (Equation 4)–(Equation 6), as well as the simulation result (Figure 2), confirm that resistance (Equation 2) and reactance (Equation 3) depend on frequency in a non-linear way.

The non-zero reactance value below and above the resonance frequency leads to unwanted effects. As presented in more detail in the following chapters, the presence of reactance causes back-propagation of the energy to the source, and in the case of the antenna, it is related to unwanted back radiation.

## 6. Experiments

Theoretical losses on the antenna presented in the above sections were observed during the experiments. As the amount of retransmitted energy was estimated to be very limited, it was decided that the simulation environment would be the best fit for verification. The amount of energy losses is extremely small and in order to properly present the backward energy propagation phenomenon, the circuit was simplified, and real components were reduced to diodes to focus only on the main problem. Using a real test bench means the presence of various wave effects such as reflection or diffraction. It would also be difficult to fully eliminate electromagnetic noise which would affect the results. Although special RF chambers are available, the precision of instruments may also affect the results of the research. Therefore, simulations in KiCad EDA were performed using the SPICE model commonly used in RFEH designs HSMS-285x diode [29]. Experiments were conducted on the model presented in Figure 3.

This simple model consists of the equivalent Thévenin circuit, which models the voltage on the receiving antenna of 50 Ω impedance. Figure 4 presents the real antenna (a), its dipole model (b), and corresponding equivalent Thévenin circuit (c).

The next stages are the LC circuit, the half-wave rectifier, and load, which represent the application module which uses the harvested energy. Series LC circuits were chosen because they increase voltage the most efficiently to the resonant frequency. A sufficiently high voltage is necessary to “pass” through the diode. As mentioned, real diode parameters are used; therefore, the real life behavior of the rectifier is modeled. However, all other electronic elements are ideal.

Model presented on Figure 3 was simulated using the SPICE simulator built in KiCad EDA. The parameters used were AC of amplitude equal to 0.69 V. The chosen frequency is 433 MHz. This frequency was chosen because it is one of the ISM (Industrial, Scientific, and Medical) bands, which means that transmission is possible without any licensing [30]. Such parameters used in the equivalent Thévenin circuit simulate energy being received via the antenna from 433 MHz as proposed in [24]. During simulation of the model presented, energy losses were observed. Some energy is expected to be anticipated in the circuitry as the imperfection of the electronic parts can cause that. However, the only SPICE model that was not ideal in this circuit was a HSMS diode. The flow of losses was analysed and, apart from the ones from the diode, it was noted that the equivalent circuit which acts as the receiving antenna gets some of the energy back from the harvesting module. Figure 5 shows the power available on the receiving antenna substitute.

Figure 5 presents the energy transfer between the antenna and the harvesting module. The time axis is shifted from zero to show the power in the steady state. Negative values represent the power output from the antenna to the harvesting module. However, it can be seen that energy is transferred not only from the source to the system, but also that the voltage source receives energy, represented by positive power, as clearly shown in Figure 6.

Once again the time axis is shifted right to show the steady state. The backward propagation of part of the energy to the receiving antenna may be the effect of the diode capacitance on the LC resonant circuit. This circuit was used to increase the voltage before the rectifier circuit. It is designed to minimise losses during energy transfer between the two parts of the circuit. The circuit operates at high frequencies, which causes the LC circuit to return part of the energy to the receiving antenna as a result of the backward propagation phenomenon. This causes the receiving antenna to re-emit the returned energy.

## 7. Solution

The conducted experiments provided a proof for the phenomenon discussed in the theoretical part but also led to the debate on the possible ways of overcoming such losses. The following assessment was conducted:LC circuit is used to increase voltage before the rectifier. This is performed to minimize energy losses during power transfer between the receiving antenna and the rectifier.LC circuit works at high frequency with different load, causing the return propagation of energy back to the receiving antenna.The internal capacity of the rectifier diode causes return propagation in a half-wave rectifier. During the non-coducting period, internal capacity causes back-propagation to the antenna.

It is inevitable to use both the LC circuit and rectifier stage in the design of a fully operable Radio Frequency Energy Harvester. Both parts are crucial for harvesting and reusing the energy propagated during wireless transmission. However, the above considerations led to the conclusion that it should be possible to minimize the backward energy propagation via a different rectifier model. The original model used the most common and easiest rectifier circuit. The half-wave rectifier requires a single diode, which is the first choice in applications where every single electronic part can cause energy losses. However, it was decided to verify how the full-wave rectifier would behave in given conditions. Therefore, a new model was proposed as in Figure 7. It uses the same component SPICE models as the original one; hence, the only difference is in rectifier architecture.

The parameters of both the model and the simulations were kept the same between both models. This approach made it possible to verify both models’ operating conditions similar to those in real environment. As the amount of energy is very little here, the module is subject to a rather uneasy operating condition. Having the ability to set the same parameters in simulation for a full-wave rectifier, the new model was subjected to the same simulation conditions as the original one. The results of the simulations are presented in Figure 8. P(V5) is the power on the equivalent voltage source. Its negative value means that energy is transferred from the antenna being here the energy source to the rest of the harvesting circuit.

As the amount of energy being discussed was very little, both models’ performance was compared. To better visualize the difference in the energy backward propagation, the results for half-wave and full-wave rectifiers were presented on a single figure as shown in Figure 9. This was only possible because the simulation was performed under the corresponding conditions. In addition, the results presented can be analyzed using built-in tools, which provides a handy way to compare the data.

Figure 9 shows that the backward propagation was significantly reduced thanks to the introduction of the full-wave rectifier model. Figure 9 is also shifted to the right to focus on the steady state of the harvesting unit. Not only is the full-wave rectifier backward energy propagation much limited in comparison to the half-wave rectifier model, but it also has much less fluctuation in the particular cycles.

## 8. Discussion

The backward energy propagation phenomenon presented in this article was observed for the first time during works on [24]. It brought attention as some of the energy was missing in the circuit. The works presented in this paper supported the initial observations. The theoretical considerations on both the antenna and the dipole impedance supported the observations and provided the background for the backward energy propagation phenomenon verification through the simulation of Radio Frequency Energy Harvesting circuits. Simulating even a model as basic as the one presented in Figure 3 provided clear evidence that some of the energy received on the antenna is coming back to it from the other parts of the harvester. Not only did it lead to further investigation, but also it proved the importance of simulation research. Having the ability to work with the mix of ideal and real models of electronic parts made it possible to identify a possible loss path of the energy in the circuitry. Eliminating losses due to the internal resistance of particular components allowed to identify the rectifier circuit as the source of the backward energy transfer phenomenon. Access to the most commonly used diode model in Radio Frequency Energy Harvesters research provided a unique possibility to recreate the real model of the rectifier stage. It was crucial to conduct simulations under the same conditions so that the results could be directly compared. Using a simulation environment made it possible to compare data in the steady state of the harvesting process. The obtained results clearly showed that the energy transferred back from the harvesting circuit to the receiving antenna was transferred back to the receiving antenna, but also demonstrated that the proposed solution with the full-wave rectifier reduces the amount of energy that is remitted. Having those initial results, it is possible to further work on the minimization of the energy that is being transferred back to the antenna during the harvesting process. Although following the whole energy transfer path is extremely difficult in conditions such as those present in Radio Frequency Energy Harvesting environment, preparing the energy loss path can be further considered.

## 9. Conclusions

Research on Radio Frequency Energy Harvesters has only recently become more rapidly available. Although the first research papers are from the 1970s in which rectennas were considered [31], this topic has been widely addressed over the last ten years. As the main goal of every harvester is to be the most efficient, researchers focus on reaching the best energy-transfer ratio, often focusing on just a single part of the circuit. However, it is the optimization of the entire model that is crucial in designing a successful harvester. Systems for harvesting energy transmitted by electromagnetic waves require a precise design. Due to the small amount of energy available to harvest, any losses negatively affect the efficiency of the system. The phenomenon of losses resulting from the capacitance of diodes, which affects their switching time and thus allows reverse current to occur in systems operating at higher frequencies, is a well-known issue. The identified and described phenomenon of reverse energy propagation to the receiving antenna is, on the other hand, rarely addressed. Therefore, further research will be conducted to describe the phenomenon and develop a method to minimise it. The approach presented in this paper provides promising results. However, the precise design of diode-based rectifiers for Radio Frequency Energy Harvesting applications is limited by manufacturing technology, which has limited possibility to improve Schottky diode parameters. It is therefore suggested that the issue is further addressed using a more sophisticated CMOS approach, which already has promising results in Radio Frequency Energy Harvesting models. CMOS designs have less internal losses and can easily be adjusted to the target frequency [32]. In addition, CMOS-based rectifiers can be put in stages to achieve wider frequency band coverage [33]. However, even CMOS-based solutions do not ensure a lossless transfer of energy. It is always some part of the energy that is lost during the harvesting process. This is why maximization of the energy available is crucial; wideband Radio Frequency Energy Harvesters can be used to do this. As was presented, area of research connected to Radio Frequency Energy Harvesting is broad and a lot is still to be worked on.

## 10. Further Works

There are at least a few possible directions for further work. The use of resonance circuits for voltage multiplication purposes in RF Energy Harvesters is one of the most interesting. This requires a precise design of the harvesting devices, but it is also promising when it comes to harvesting energy dissipated in wider frequency range. This is an important aspect, as radio transmissions currently use wider bands than previously. As more and more transmissions are conducted wirelessly, the spectrum is much more densely packed. As for the initial harvesters it was enough to collect energy from center frequency peak energy transmission such as this connected to FM radio broadcasting, now much more transmission is using wide bands to provide higher data volume transfer. Research on energy backward transmission minimization is also important in real circuits. This is why studies on the RF chamber are necessary. Such a chamber has to meet at least a few requirements. First, it has to isolate the circuit and the measurement instrument placed inside it from the external environment. As wireless communication is almost ubiquitous, it is crucial to reduce the environment noise as much as possible. The second requirement is to minimise the internal wave effects, such as diffraction and reflection. The harvesting unit should be under the impact of an isolated energy source that reflects the real conditions as much as possible. The use of standardized and postulated simulations and measurements is crucial to further compare research data. Currently, data provided in various research is almost incomparable. Every single Radio Frequency Harvester is verified under different conditions. For simulations, the conditions are often unknown or differ. For the real devices, the measurement environment is rarely isolated. There are many review papers on Radio Frequency Energy Harvester [34], but the results are not normalized. Therefore, it is impossible to state which designs are the most efficient. Rapid development of wireless communication creates the unique possibility to propose a standard for wireless charging using RF communication. Using Private Wireless Networks (PWN) [35] in industry gives a chance to power the Internet of Things sensors in a controlled environment. Last but not least, the Radio Frequency Energy Harvesting can be considered in the extraterrestrial use cases. The rapidly increasing number of nano- and micro-satellites that are communicating with each other can be seen as a swarm of Radio Frequency Energy Harvesters. Those small devices are constantly receiving and transferring data. What is more, the non-terrestrial environment has less attenuating characteristics, meaning that more energy can reach the target. Even further, the plans for colonization of the Solar System planets such as Mars can consider Radio Frequency Energy Harvesters as one of the critical infrastructure systems. RF Energy Harvesters are complex but really promising devices that can provide power for a lot of applications. Further works in this matter will be crucial for achieving not only sustainable but also cheap energy in the future.

## Figures and Tables

**Figure 1 micromachines-16-01187-f001:**
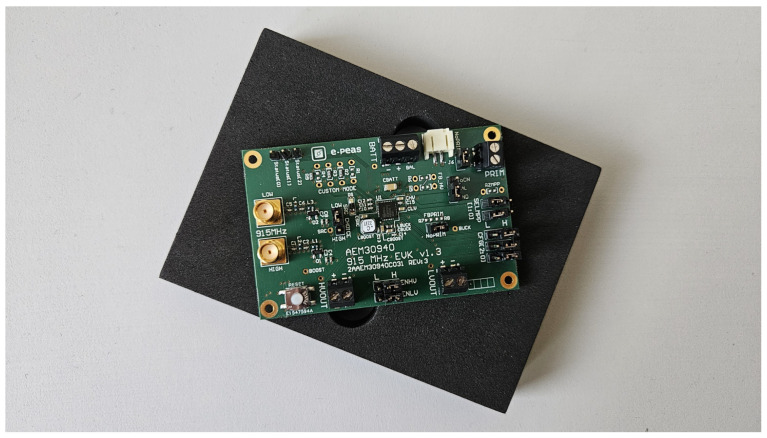
e-peas AEM30940 RF Energy Harvester evaluation board.

**Figure 2 micromachines-16-01187-f002:**
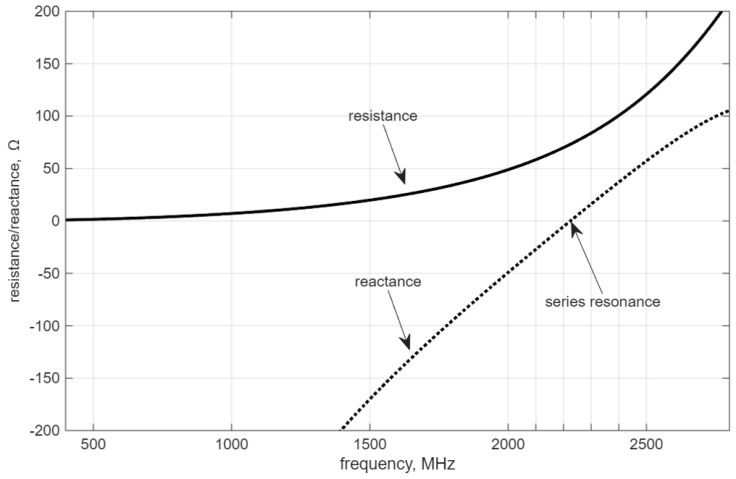
Numeric calculation of the dipole antenna impedance.

**Figure 3 micromachines-16-01187-f003:**
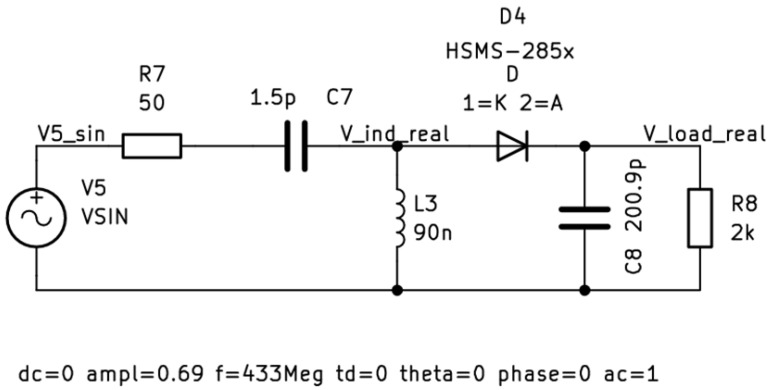
Simulation model with a resonance circuit, on which the energy losses were observed.

**Figure 4 micromachines-16-01187-f004:**
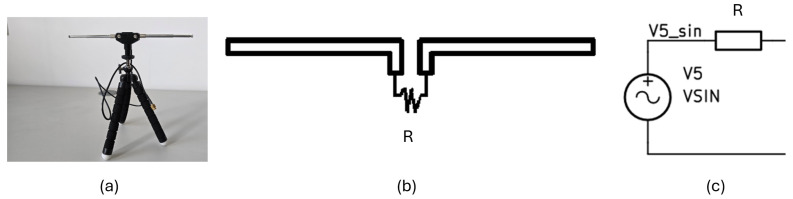
Antenna model used in simulation. (**a**) Real antenna, (**b**) dipole model, (**c**) equivalent Thévenin circuit.

**Figure 5 micromachines-16-01187-f005:**
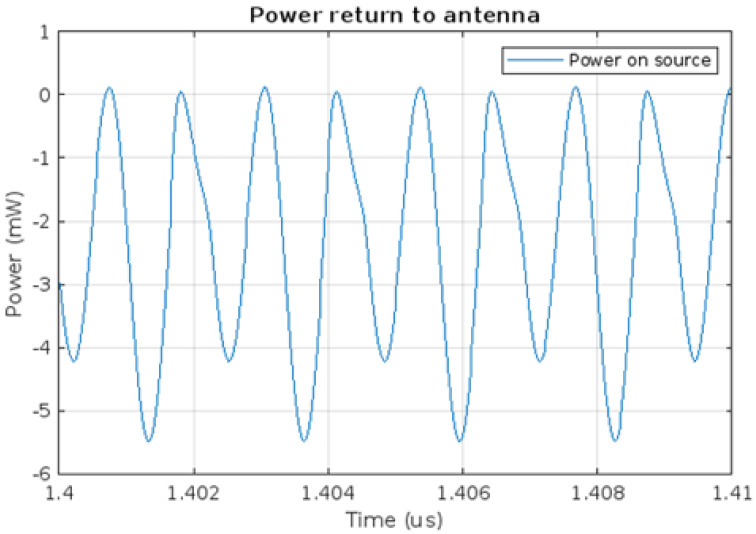
Power transfer at the voltage source in the circuit.

**Figure 6 micromachines-16-01187-f006:**
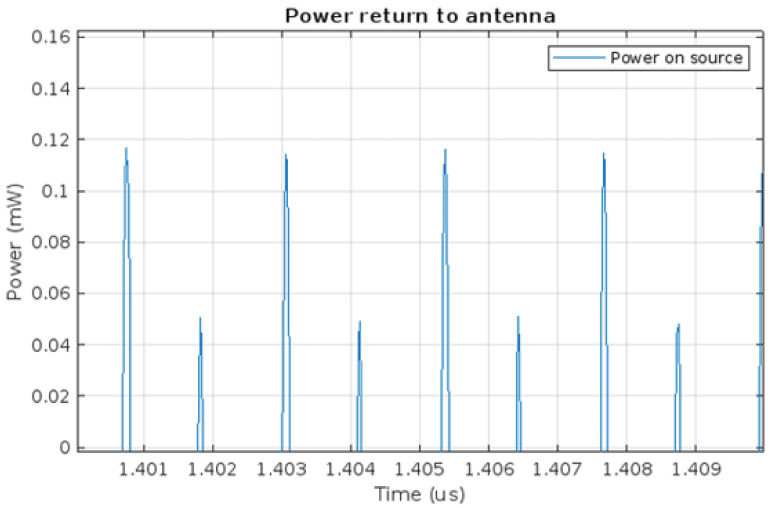
Energy returning to the voltage source–power supplied.

**Figure 7 micromachines-16-01187-f007:**
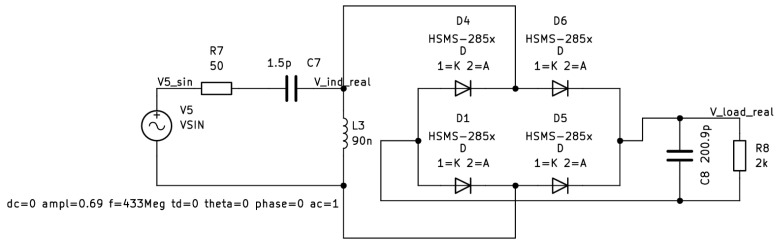
Simulation model with a resonance circuit and a full-wave rectifier.

**Figure 8 micromachines-16-01187-f008:**
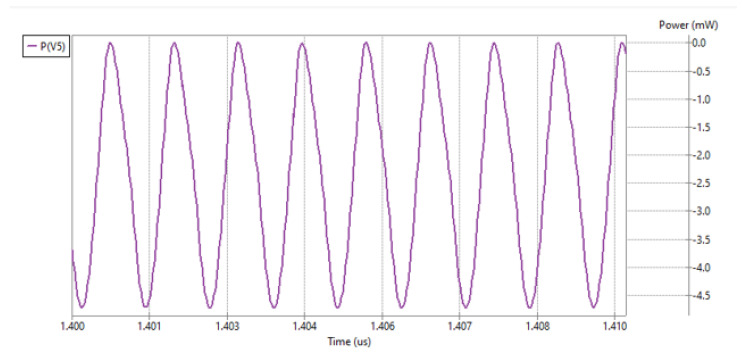
Energy returning to the voltage source—power supplied. Results for full-wave rectifier model.

**Figure 9 micromachines-16-01187-f009:**
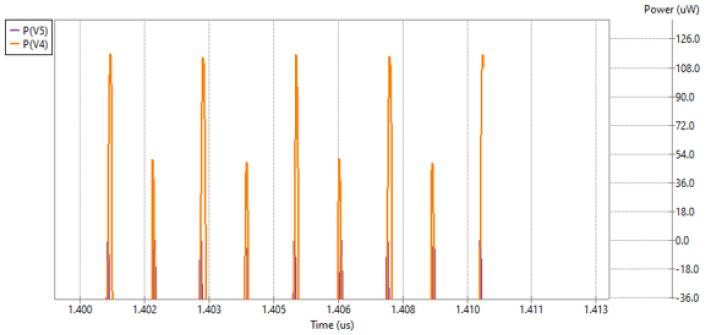
Power return to antenna with half- (V4) and full-wave (V5) rectifier.

## Data Availability

The data presented in this study are available on request from the corresponding authors.

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
