# Peer review of "Backward Energy Transmission in Resonant RF Energy Harvesters"

_micromachines, 2025, doi:10.3390/mi16101187_

Round 1
Reviewer 1 Report
Comments and Suggestions for Authors
The paper discussed the reverse energy propagation to the receiwing antenan, which is vital for energy harvesting efficiency of RFEH. There are some issues need to be addressed:
1) To aviod the power loss, the impedance matching network is required to reduce transmission loss form the receiving antenna. How to achieve the matching in your study? The author claims that there are still energy loss, how about the loss path? Please provide a comparison among different loss paths.
2) After integrating full-wave rectifier model, the backward energy is significantly reduced compared with half-wave rectifier. Please explain the reason.
Author Response
Dear reviewer,
Thank you for your work and your comments. Those brought our attention to the parts of our article that need some improvements. Please find all of changes included in the latest version of the manuscript and below we've prepared answers to your comments. We hope that all the improvements we have made will ensure you that this manuscript is worth publishing in its current form.
Comment 1) To aviod the power loss, the impedance matching network is required to reduce transmission loss form the receiving antenna. How to achieve the matching in your study? The author claims that there are still energy loss, how about the loss path? Please provide a comparison among different loss paths.
Response: It's indeed a matter of impedance matching network to minimise the energy loss in transfer between receiving antenna and the rest of the circuit. There are a lot of approaches to that proposed in the literature and those added and briefly discussed in the revised version. However, as main goal of this article was to present the energy backward propagation phenomenon, the deep discussion of impedance matching network wasn't included as it is a matter of further study to propose the overall solution for optimized RF Energy Harvester.
When it comes to the loss path, the loss coming from backward propagation was presented on Fig.4 and Fig.6 and discussion of those results was extended in the revised version. Description of different path losses is interesting area and somehow was discussed by authors in other article (10.24425/aee.2025.154991). However, further discussion of path losses is a good idea for next article.
Comment 2) After integrating full-wave rectifier model, the backward energy is significantly reduced compared with half-wave rectifier. Please explain the reason.
Response: The explanation was added in the revised version. The reason for backward energy transmission reduction is that the whole wave is received in the rectifier circuit and processed there when full wave model is used. Energy is therefore not blocked and retransmitted back to the antenna. The use of proper impedance matching network mentioned in the previous comments should reduce the energy backward propagation even more. One of the approaches considered by authors for further works is resonance effect to increase the voltage available in the circuit to use it to power up further circuit stages.
Reviewer 2 Report
Comments and Suggestions for Authors
The authors need to address the below issues before publication.
- Please carefully double check the manuscript writing and correct the errors.
- The introduction is too short, more literature review required.
- The authors should do a benchmarking between resonant RF energy harvesters and others, and elaborate why they did research do RF energy harvester.
- Please clearly point out the meaning of symbols in equations.
- Schematics of antenna, dipole and the RF energy harvester, and the corresponding real images.
- The authors considered the LC circuits, how about RC、LR、LRC circuits?
- More results and discussion required.
- More references required.
Author Response
Dear reviewer,
Thank you for your work and your comments. Those brought our attention to the parts of our article that need some improvements. Please find all of changes included in the latest version of the manuscript and below we've prepared answers to your comments. We hope that all the improvements we have made will ensure you that this manuscript is worth publishing in its current form.
- Please carefully double check the manuscript writing and correct the errors.
- We have double checked the manuscript and do whatever possible to eliminate all mistakes in writing.
- The introduction is too short, more literature review required.
- Introduction was extended and more literature was added. We introduced extended topic presentation and cited core literature to help readers find more about not only the phenomenon described in this paper but also the whole RF Energy Harvesting topic.
- The authors should do a benchmarking between resonant RF energy harvesters and others, and elaborate why they did research do RF energy harvester.
- Authors research focuses on RF energy harvesters as those can reuse energy that is already propagated in the environment via telecommunication and other wireless systems. Resonant circuits were chosen because they are the most effective at increasing the voltage at the resonant frequency. Harvesting this energy can be used to power up passive sensors without need for batteries or other external power sources.
- Please clearly point out the meaning of symbols in equations.
- Meaning of symbols in equations were explicitly described in the revised version.
- Schematics of antenna, dipole and the RF energy harvester, and the corresponding real images.
- Antenna and dipole were not subject of this research and therefore those are not considered in this article. This part of the circuit is modeled using Thevenin's voltage source, which is commonly used to replace receiving antenna. Using proper parameters of voltage source, energy received on the antenna is modeled. RF harvester model was provided and further more general RF harvester scheme was added. Real images are not included as this is purely simulational paper due to very small amount of energy observed that would not be observable in real life environment due to background noise. Further research is planned to use special echo-cancelling chamber that is currently under construction.
- The authors considered the LC circuits, how about RC、LR、LRC circuits?
- Series LC circuits were chosen because they increase voltage most efficiently to the resonant frequency. A sufficiently high voltage is necessary to "pass" through the diode. The other circuits mentioned cannot provide equally satisfactory results.
- More results and discussion required.
- Results description was further developed and discussion was extended to cover both current state of research and plans for further works.
- More references required.
- Article was extended with additional references and citation to back presented theory and data.
Reviewer 3 Report
Comments and Suggestions for Authors
In this article, the authors address the issue of backward energy transmission in resonant circuits using radio frequency energy harvesting technology. This topic is rarely addressed in the literature, therefore it is significant that the authors clearly present the essence of the problem, develop a mathematical model of the device, and identify the source of unwanted backward emission.
The entire presentation is coherent and clear, and the proposed method using a full-wave rectifier provides a practical solution for reducing this effect. However, the limitation of the analysis to simulations using ideal components (except diodes) and the lack of experimental data supporting the obtained results raises concerns.
The article is written in a clear manner, but expanding the analysis and verifying the results on real systems would significantly enhance the scientific value of the work.
Author Response
Dear reviewer,
Thank you for your work and your comments. Those brought our attention to the parts of our article that need some improvements. Please find all of changes included in the latest version of the manuscript and below we've prepared answers to your comments. We hope that all the improvements we have made will ensure you that this manuscript is worth publishing in its current form.
In this article, the authors address the issue of backward energy transmission in resonant circuits using radio frequency energy harvesting technology. This topic is rarely addressed in the literature, therefore it is significant that the authors clearly present the essence of the problem, develop a mathematical model of the device, and identify the source of unwanted backward emission.
Response: Thank you for your review! The introduction was extended and references to the more introductory articles were added to the revised version to even more extend the background for RF Energy Harvesters.
The entire presentation is coherent and clear, and the proposed method using a full-wave rectifier provides a practical solution for reducing this effect. However, the limitation of the analysis to simulations using ideal components (except diodes) and the lack of experimental data supporting the obtained results raises concerns.
Response: In the revised version of the article, the explanation why study is based on ideal components is extended. The amount of energy losses is extremaly small and in order to properly present the backward energy propagation phenomenon, the circuit was simplified and real components were reduced to diodes to focus only on the articles main problem. Further analysis of the system will be provided in the next works which will consider impedance matching network analysis and resonance effect used to minimise losses even further. Current article aim was to present the problem and provide initial conteractions to prevent from it. This was further discussed in the revised version of the document.
The article is written in a clear manner, but expanding the analysis and verifying the results on real systems would significantly enhance the scientific value of the work.
Response: Providing the experimental data based on the real harvester may not provide any reasonable results due to the amount of interferences and background noise. Authors are working on the echo-cancelling chamber to conduct research on the real harvester circuits but even there it may be difficult to observe such small energy levels as presented in this article. This discussion was also added to the revised version of the article.
Round 2
Reviewer 1 Report
Comments and Suggestions for Authors
All my concerns are well-addressed.
Author Response
Dear Reviewer,
Thank you for reviewing our article.
We wish you all the best in your own research,
Authors
Reviewer 2 Report
Comments and Suggestions for Authors
The authors need to address the below issues before publication.
- Please carefully double check the manuscript writing and correct the errors.
- Schematics of antenna, dipole and the RF energy harvester, and the corresponding real images.
- The authors considered the LC circuits, how about RC、LR、LRC circuits?
Author Response
Dear Reviewer,
We are really appreciating your work and comments you are sharing with us. We have introduced a view changes that hopefully will resolve issues that you have pointed out. Below, please find our responses to every single of your comments.
- Please carefully double check the manuscript writing and correct the errors.
- Authors have gone through the article once again, correcting number of grammar, punctuation and style issues. Hopefully all errors were caught up and cleaned.
- Schematics of antenna, dipole and the RF energy harvester, and the corresponding real images.
- Real image of antenna, corresponding dipole model and equivalent Thévenin circuit were presented in Fig. 3 to clearly show how simulation models the antenna's received voltage. Real RF energy harvester that is used in other research but presents the general idea of RFEHs evaluation boards was presented on Fig. 1. Authors are planning to prepare their own RFEH circuit as research continues. Then, a separate manuscript will be prepared.
- The authors considered the LC circuits, how about RC、LR、LRC circuits?
- We appreciate the reviewer’s comment and agree that RC, LR, and LRC circuits can also be analyzed in similar contexts. However, the focus on the LC configuration was intentional. The series LC circuit exhibits a pronounced resonant behavior that enables significant voltage amplification across the capacitor at the resonance frequency. This feature is essential for achieving the voltage level required to drive the nonlinear diode element in our setup. In contrast, RC and LR circuits do not provide resonance-based voltage magnification, while LRC circuits, although resonant, introduce additional damping due to the resistive component, which lowers the achievable voltage gain. For these reasons, the LC circuit was selected as the most effective configuration for the intended application.
Once again, thank you for your work and we wish you all the best in your own research,
Authors